# Association between Functional Parameters of Coagulation and Conventional Coagulation Tests in the Setting of Fluid Resuscitation with Balanced Crystalloid or Gelatine: A Secondary Analysis of an In Vivo Prospective Randomized Crossover Study

**DOI:** 10.3390/jcm11144065

**Published:** 2022-07-14

**Authors:** Agnieszka Wiórek, Piotr K. Mazur, Bożena Niemiec, Łukasz J. Krzych

**Affiliations:** 1Department of Anaesthesiology and Intensive Care, Faculty of Medical Sciences in Katowice, Medical University of Silesia, 40-572 Katowice, Poland; lkrzych@sum.edu.pl; 2Department of Cardiovascular Surgery, Mayo Clinic, Rochester, MN 55905, USA; piotr.k.mazur@gmail.com; 3Department of Cardiovascular Surgery and Transplantology, Institute of Cardiology, Jagiellonian University Medical College, 31-202 Cracow, Poland; 4Central Laboratory, University Clinical Centre of the Medical University of Silesia, 40-572 Katowice, Poland; bniemiec@uck.pl

**Keywords:** perioperative medicine, fluid resuscitation, fluid therapy, point-of-care testing, coagulation and fibrinolysis, rotational thromboelastometry

## Abstract

Functional point-of-care tests (POCTs) have evolved into useful tools for diagnosing disorders of blood coagulation and fibrinolysis. We aimed to describe the in vivo association between standard and functional parameters of coagulation and fibrinolysis in the setting of acute hemodilution induced by an infusion of balanced crystalloid or synthetic gelatine solutions. This prospective randomized crossover in vivo study included healthy male volunteers aged 18–30 years. Enrolled participants were randomly assigned to receive either the Optilyte^®^ or Geloplasma^®^ infusion. Laboratory analysis included conventional coagulation parameters and rotational thromboelastometry (ROTEM) assays. A total of 25 healthy Caucasian males were included. ROTEM viscoelastic assays presented moderate to strong correlations with conventional coagulation tests, regardless of the fluid type utilized. Irrespectively of the extent of hemodilution, significant correlations remained unaffected. The strongest associations were found between the ROTEM clot formation and clot strength and the fibrinogen concentration and platelet count, and between the ROTEM clotting time and the APTT and PT. This in vivo experimental study in healthy male volunteers demonstrated that ROTEM may be used as a credible alternative to standard laboratory tests to assess blood coagulation and fibrinolysis in the setting of fluid resuscitation with both crystalloid and colloid solutions. The study was registered online in the ClinicalTrials.gov database (NCT05148650).

## 1. Introduction

Functional point-of-care tests (POCTs) have evolved into a useful tool for diagnosing disorders of blood coagulation and fibrinolysis [1,2,3]. Using POCTs instead of conventional coagulation studies to diagnose the cause of coagulopathy shortens the time to treatment initiation and results in improved survival, as has been shown in the American Pragmatic Randomized Optimal Platelet and Plasma Ratios (PROPPR) studies [4,5]. However, the use of POCTs is not considered the gold standard of care due to a lack of evidence [6,7].

The standard laboratory tests, e.g., prothrombin time (PT), activated partial thromboplastin time (APTT), and fibrinogen concentration, remain recommended for the treatment of coagulation disorders [8], but in the acute setting of hemorrhage followed by aggressive fluid resuscitation, POCTs provide a clinician with information in a timelier fashion. Data on the correlations between classic tests and POCTs in the setting of fluid resuscitation are lacking. Examples of published results tackle the issue, for instance in the context of extracorporeal support, liver transplantation, and pediatric surgery, with mixed conclusions [9,10,11,12]. POCTs used to globally evaluate hemostasis include rotational thromboelastometry (ROTEM), thromboelastography (TEG), impedance platelet aggregometry, and activated clotting time (ACT), among others. We chose to focus on ROTEM, as this system operates on four independent channels that allow for the simultaneous use of specific assays to compare different aspects of coagulation dynamics.

In this randomized study on healthy volunteers, we aimed to describe the in vivo association between standard and functional parameters of coagulation and fibrinolysis, that is, to evaluate the statistical correlation between these parameters, taking into account the degree of dilution associated with the fluid infusion in the setting of acute hemodilution induced by an infusion of balanced crystalloid or synthetic gelatine solutions. We hypothesized that independently of the infused fluid type and degree of induced hemodilution, the results of ROTEM are well correlated with the classic coagulation studies.

## 2. Materials and Methods

The study was registered online in the ClinicalTrials.gov database (NCT05148650), and the primary data, including the full methodology, were previously described and published [13].

### 2.1. Study Participants

This prospective randomized crossover study was designed to include healthy male volunteers aged 18–30 years in the American Society of Anesthesiologists physical status class I. All participants were recruited by informational flyers distributed throughout the hospital and among academic staff members. The exclusion criteria were female sex, blood type O, a positive history of any acute diseases in the last four weeks, chronic diseases, any diagnosed hemostatic disorders, history of anticoagulation, known bleeding diathesis, and any pharmacotherapy in the previous week. Blood type was determined with medical records in all candidates. Participants were informed about the prohibition of alcohol intake, excessive exercise, and stress on the day before blood sampling. There was no reimbursement of costs or financial support for participating in the project. The patients’ flow diagram is shown in Figure 1. All demographic and medical data were recorded prospectively.

The study was approved by the Ethics Committee of the Medical University of Silesia in Katowice, Poland (KNW/0022/KB1/159/II/15/16/18/19). Written informed consent was obtained from all participants. The Consolidated Standards of Reporting Trials (CONSORT) Statement was applied for appropriate data reporting.

### 2.2. Study Design and Interventions

Infusions included 20 mL/kg of a balanced crystalloid solution (Optilyte^®^, Fresenius Kabi, Bad Homburg, Germany) or 20 mL/kg of gelatine 26.500 Da (Geloplasma^®^, Fresenius Kabi, Bad Homburg, Germany) in a random order over a period of 2 weeks (washout period). The 14-day washout period was established based on the available literature concerning early experiments with in vivo changes in coagulation after fluid infusion [14]. The infusions were performed through an 18 G intravenous cannula inserted into an antecubital vein in the non-dominant limb at a rate of 1000 mL/h. Based on this infusion rate and depending on the amount of fluid to be infused (calculated from the participant’s weight), the infusion took between 90 and 120 min. The first blood sample was collected before the start of the infusion, and the second immediately after the infusion of the test solution.

### 2.3. Randomization

The enrolled participants were randomly assigned at a 1:1 ratio to receive either the Optilyte^®^ or Geloplasma^®^ infusion. Before the onset of the study, opaque envelopes containing equalized numbers of cards indicating the type of solution to be infused (Optilyte^®^, *n* = 13 or Geloplasma^®^, *n* = 12) were sealed, shuffled, and numbered from 1 to 25. The principal investigator (AW) enrolled all participants. Consecutive participants received numbers corresponding to the consecutive numbers on the envelopes. The investigator responsible for administering the infusions opened the envelopes just before entering the lab and was not blinded to the test solutions. All participants received the infusions based on the same principle of dose calculation and infusion rate. Participants and the Central Laboratory team performing the standard laboratory tests were blinded to the test solution assignments. The technician performing the ROTEM assays was also blinded to the type of infused fluid. After a washout period of 14 days, the participants returned to receive the second type of fluid using the exact same method and volume as calculated for the first infusion.

### 2.4. Laboratory Investigations

Blood samples were collected from an antecubital vein with minimal stasis between 3 and 5 PM on the appointment day following verification of the adherence to the study conditions. Two blood samples, 12.5 mL each, were obtained just before and immediately after the fluid infusion using a vacuum system (BD Vacutainer^®^, Franklin Lakes, NJ, USA), from a separate venipuncture on a contralateral extremity with minimal stasis. The samples were collected in containers with ethylenediaminetetraacetic acid (EDTA) and 3.2% buffered sodium citrate.

Standard laboratory tests included: fibrinogen concentration, D-dimer concentration, APTT, PT with the calculation of the international normalized ratio (INR), hematocrit, hemoglobin concentration, and platelet count (PLT). The fibrinogen concentration was assessed based on the Clauss method for the quantitative determination of fibrinogen, using thrombin to measure the fibrinogen in human citrated plasma on the IL Coagulation System [15]. The degree of hemodilution was expressed as a percentage of the difference in the hematocrit value before infusion and after infusion, divided by the hematocrit value before infusion [(Ht_before_ − Ht_after_)/Ht_before_ × 100%].

Rotational thromboelastometry was carried out using a ROTEM delta analyzer (Tem Innovations GmbH, Munich, Germany) following the manufacturer’s instructions. The detailed meanings of particular ROTEM parameters were widely described in the companion study by the same scientific team published as part of an extensive research project [13]. The assays were allowed to run for 60 min immediately after blood sampling. The ROTEM analyzer was checked and prepped by a trained technician directly before the admission of the participant and available in the same room where the infusions took place. Three ROTEM assays were run simultaneously, INTEM, EXTEM, and FIBTEM. The intrinsic pathway (INTEM) utilizes an ellagic acid as the contact activator, which leads to the activation of the intrinsic coagulation pathway and estimates the functional ability and availability of factors XII, XI, IX, and VIII, along with factors of the common pathway. The (extrinsic pathway (EXTEM) assay includes tissue factor that leads to the initiation of the extrinsic coagulation pathway sensitive to the activity of factor VII and common pathway factors. The fibrinogen (FIBTEM) assay combines the EXTEM base for clot assessment after the addition of cytochalasin D, which is a platelet inhibitor, allowing for the isolated assessment of the fibrin contribution to the clot. [16]. All ROTEM analyses were performed by the same technician. The parameters of interest measured in the three assays were the clotting time (CT), clot formation time (CFT), alpha angle (AA), the amplitude at different time points (minutes) (A10, A20), maximum clot firmness (MCF), and maximum lysis (ML) [17]. The maximum clot elasticity (MCE) values for the EXTEM and FIBTEM assays were calculated with the following formula: MCE = 100 × MCF/100 − MCF. The assessment of the platelet contribution to clot strength was measured according to the formula: ΔMCE = MCE_EXTEM_ − MCE_FIBTEM_.

No protocol violations were recorded after the initiation of the study. A summarized graphical illustration of the study protocol is depicted in Figure 2.

There were no losses of participants or deviations from protocol throughout the course of the study. The primary endpoint was coagulation and fibrinolysis impairment after the infusion of balanced crystalloid and balanced colloid.

### 2.5. Statistical Analysis

Statistical analysis was performed using MedCalc v.18 software (MedCalc Software, Ostend, Belgium). No a priori power or sample calculation was performed. Quantitative variables were depicted using medians and interquartile ranges (IQR). The Shapiro–Wilk test was used to verify their distributions. Qualitative variables were described using frequencies and percentages. The between-group differences between undiluted and diluted samples were analyzed using the Friedman test or the paired samples t-test depending on the distribution normality. Post-hoc comparisons were performed with the use of the Scheffe test or Conover post-hoc analysis when appropriate. The association between the standard laboratory tests and the individual functional tests of coagulation for the pre-dilution and post-dilution samples was assessed with univariate analysis using the Spearman rank coefficient of correlation or Pearson’s correlation coefficient (R), depending on the variables’ distribution, and verified by multiple regression, where the ROTEM findings were considered dependent variables, and classical laboratory tests and the degree of hemodilution were independent variables. The correlations between parameters matched in pre-dilution/pre-dilution or post-dilution/post-dilution combinations were examined, i.e., the conventional coagulation parameter measured before crystalloid infusion was correlated with the respective ROTEM parameter before crystalloid infusion; the same applies to the results after infusion. Likewise, the results acquired for the conventional tests before colloid infusion were correlated with the ROTEM parameters before infusion, and the conventional tests acquired post-dilution were correlated with the ROTEM post-dilution results. A *p*-value of < 0.05 is considered significant. The Bonferroni correction was applied for multiple comparisons (corrected *p*-value of 0.00125).

## 3. Results

Forty individuals were screened for eligibility between February 2021 and May 2021, and 25 healthy Caucasian males were included. The median age of the participants was 25 (23–29) years, and the median BMI was 23.9 (22.5–26.7) kg/m^2^. The most frequent blood type was A Rh-positive (*n* = 13, 52%), followed by B Rh-positive (*n* = 8, 32%), A Rh-negative (*n* = 2, 8%), and B Rh-negative (*n* = 2, 8%). The median volume of both infused fluids was 1600 (1430–1700) mL.

### 3.1. Hemodilution

Both fluids caused expected hemodilution. For the crystalloid, the median hematocrit dropped from 43.2% (41.9–45.2) before infusion to 39.8% (39.2–41.5) after infusion (*p* < 0.00125). For the colloid, it dropped from 43.5% (41.2–45.4) to 36.6% (33.8–37.4), (*p* < 0.00125). Colloid infusion caused more pronounced hemodilution compared to the crystalloid solution (*p* < 0.00125). More detailed results, including the values of standard laboratory tests and ROTEM tests, with a full description of the laboratory analysis, were previously published [13].

### 3.2. INTEM/EXTEM Clotting Time

There was a strong positive correlation between the APTT and INTEM CT after both the crystalloid and colloid infusions (Table 1). These associations were confirmed in a multivariate analysis (Appendix A). There were individual associations between the EXTEM CT and PT after the infusions of the crystalloid and colloid (Table 1); however, multivariate analysis confirmed the effect only for post-crystalloid dilution (Appendix A).

### 3.3. Association with INTEM/EXTEM Alpha Angle

The INTEM AA time was correlated with the post-dilution fibrinogen concentration and the number of platelets (Table 2). These associations were confirmed with multiple regression analysis (Appendix A). The EXTEM AA was correlated with the fibrinogen concentrations for all diluted samples (Table 2). These results were confirmed with multivariate analysis (Appendix A).

### 3.4. Association with INTEM/EXTEM Clot Formation Time

The INTEM CFT was significantly negatively correlated with the fibrinogen concentrations for both diluted samples and platelet levels for both solutions (Table 3). These associations were also confirmed by regression analyses (Appendix A). The EXTEM CFT was negatively correlated only with the fibrinogen concentrations (Table 3). Regression analyses verified these findings (Appendix A).

### 3.5. Association with INTEM/EXTEM Amplitude at 10 Minutes

There were correlations between the INTEMA10 and fibrinogen concentrations for both diluted blood samples and a correlation between the INTEMA10 and platelets for the post-Geloplasma samples (Table 4), confirmed with multiple regression analysis (Appendix A). The EXTEMA10 was correlated with the fibrinogen concentrations (Table 4), which was confirmed with multivariate analysis (Appendix A).

### 3.6. Association with INTEM/EXTEM Maximum Clot Firmness

All correlations between the INTEM MCF or EXTEM MCF and the fibrinogen concentrations were statistically significant and maintained statistical importance in the regression analyses. The MCF was significantly correlated with the number of platelets in both diluted samples, which was confirmed in multiple regression analysis. There were singular positive correlations between the INTEM MCF and D-dimer concentrations, but they were not confirmed in multiple regression analysis (Table 5 and Appendix A).

### 3.7. Association with FIBTEM Parameters

The FIBTEM MCF was positively correlated with the fibrinogen concentrations in all samples, with a correlation with the FIBTEM CT for the diluted blood samples (Table 6).

## 4. Discussion

This prospective randomized crossover study on healthy volunteers demonstrates, in the experimental in vivo setting, the associations between the standard and functional parameters of coagulation and fibrinolysis obtained with point-of-care testing in the event of coagulopathy induced by the infusion of balanced crystalloid or synthetic gelatine solutions. We found that (1) ROTEM viscoelastic assays presented moderate to strong correlations with the conventional coagulation tests, regardless of the character of fluid utilized for fluid resuscitation; (2) irrespectively of the extent of hemodilution calculated for the dose of received fluid, significant correlations (verified by multiple regression analysis) remained unaffected; (3) the strongest associations were found between the ROTEM clot formation and clot strength and the conventional Clauss method for fibrinogen concentration, platelet count, APTT, and PT.

### 4.1. Dilutional Coagulopathy Detection

Past research attempted to verify if conventional laboratory coagulation tests should still be considered the gold standard or if viscoelastic tests could more accurately and earlier detect coagulopathy caused by fluid resuscitation with crystalloids or colloids [18]. The potential superiority of viscoelastic tests over plasma coagulation tests lies in the immediate availability and assessment of multiple aspects of clot formation based on whole blood, not plasma alone [19]. The methods of functional coagulation assessment are used to guide the transfusion of blood products and/or coagulation factor concentrates for a targeted correction of coagulopathy [20]. Fibrinogen is one of the strongest predictors of bleeding and transfusion requirement in the intraoperative setting during surgeries involving blood loss and requiring fluid therapy [21]. Conventional testing for fibrinogen with the Clauss method is measured in plasma and, as such, is not affected by platelet activity. On the contrary, whole-blood viscoelastic tests are affected by the platelet component, and insufficient platelet inactivation can cause falsely elevated readings of fibrinogen contribution to clot formation [22]. This shortcoming can, to some extent, be limited by calculating the ROTEM parameter of the MCE and assessing the platelet contribution to the clot by the equation of ΔMCE = MCE_EXTEM_ − MCE_FIBTEM_ [23]. However, the better method of verifying the platelets’ functionality is to perform separate additional testing through dedicated equipment, such as multiple electrode aggregometry and light transmission aggregometry (for instance, Multiplate or PFA200) [24,25].

Observations derived from our study seem to suggest the reliability of faster ROTEM results in comparison to conventional coagulation tests. This should be considered useful, particularly in intraoperative and emergency settings, regardless of facility-specific algorithms for managing massive bleeding, initial fluid load reflected by various degrees of hemodilution, and choice of fluid for resuscitation, which may favor the use of either crystalloids, colloids, or a mixed regime for the supplementation of lost volume [26,27]. The issue of the fluids’ volume effect holds significance in fluid resuscitation management, and there is still room for new observations, with reports investigating the physiological basis of longer intravascular volume maintenance after the infusion of colloids [28].

### 4.2. Association between Methods

Several studies have investigated the correlation of ROTEM with standard coagulation tests, with mixed results. We observed moderate convergence concerning one of the most utilized parameters of coagulation, that is, the clotting time, especially in the perioperative setting. It seems that the clotting time (CT) acquired with both EXTEM and INTEM assays corresponded with its respective counterparts (PT and APTT) [12]. However, in our study, the correlations were stronger for the APTT/INTEM CT and more pronounced in the multivariate analysis.

In a study by Faved et al., a moderate to good positive predictive value for intraoperative bleeding was observed for determining the need to transfuse packed red blood cells, fresh frozen plasma, or platelet concentrates based on ROTEM results with AUC values between 0.73 to 0.90 [10]. However, the authors addressed the issue that the PT and APTT cannot be used interchangeably with the ROTEM CTs, and PT and APTT values may overestimate the need for transfusion therapy [10]. As mentioned above, a comparison between ROTEM and conventional tests of fibrinogen and platelet concentration also reported sufficient agreement between the results in an in vivo setting, which is relevant when considering fibrinogen as the key coagulation factor [29]. Based on the observations of the correlation between viscoelastic and conventional coagulation tests, there is a trend for trusting the viscoelastic methods in the POC setting during goal-directed, personalized fluid resuscitation [30], concerning the coagulopathies developing in patients on chronic dialyses [31] or recently in the identification and risk stratification of thromboembolic events in patients suffering from COVID-19 [32,33]. Importantly, not all studies have taken into account blood dilution in their investigations.

### 4.3. Study Limitations

The study has some limitations. Firstly, the cohort was small, and no a priori sample size calculation was done. We based the study group size on the available published literature with similar study methodology [30,31]. Moreover, to increase the reliability of our results, we performed additional a posteriori power calculations regarding our study’s sample size for the study group. We discovered that we would require a minimum of 22 participants to confirm a significant correlation of R = 0.65 with an alpha of <0.01 and a beta of 0.20. We also performed a sample size calculation to verify the between-group differences of the hemodilution effect. We discovered that we would require at least 12 pairs to establish the significance of our hemodilution degree with an alpha of <0.0001 and a beta of 0.20. Therefore, we acknowledge that our sample size is small, but it is sufficient to draw conclusions, and our study was in no way underpowered. We included more than a minimal number of subjects calculated for both statistical purposes, even considering the Bonferroni correction applied for statistical analysis.

Furthermore, the findings in healthy young volunteers, even in an in vivo setting, may differ from those among the patients treated due to massive bleeding, hypothermia, and coagulopathy, which occur in a real clinical scenario. Secondly, we also did not include follow-up data covering the time-dependent evolution of detected coagulation impairment due to selection criteria adopted for our cohort. The inclusion of participants was based on their declaration of no known comorbidities, including the minimal risk of any kidney disorders that could cause the prolonged persistence of the hemodilution effect on coagulation or the existence of any bleeding diathesis. However, the cohort was precisely chosen to avoid significant potential confounders that could otherwise influence the differences in parameters caused exclusively by intravenous fluid administration, such as the “lethal diamond” of hypothermia, acidosis, coagulopathy, and hypocalcemia [34,35]. The degree of induced dilution remained safe for the participants. Therefore, it may have been insufficient to report further dilution-induced coagulation abnormalities even with the ROTEM analysis. ROTEM results might not always correlate with clinical signs of bleeding, as there is no blood flow or interactions with endothelium that affect coagulation in vivo. Finally, a gender criterion was implemented for patients’ exclusion, which might impact the external validity of the results for female patients. Due to possible additional blood loss associated with menstruation and the proven influence of hormonal changes on the coagulation process, as well as the female gender being a risk factor for bleeding in clinical practice in cardiac surgery procedures, women were intentionally excluded. Patients with blood type O were also excluded, as they may have genetically lower plasma von Willebrand factor levels than those with non-O blood, increasing their risk of hemorrhage, which could act as a confounding variable in the study environment, although this does not limit the clinical use of ROTEM for type O blood [36]. Lastly, there seems to be an unexplained lack of parallelism for the correlations before and after fluid infusions seen for some ROTEM parameters; however, we evaluated our data and ruled out any pre-analytic and analytic errors that could impact our calculations.

## 5. Conclusions

Based on moderate to strong correlations observed between conventional coagulation tests and ROTEM parameters, after adjusting for the degree of hemodilution, we can conclude that in healthy male volunteers, ROTEM may be used as a credible alternative to standard laboratory tests to assess blood coagulation and fibrinolysis in the setting of fluid resuscitation with both crystalloid and colloid solutions.

## Figures and Tables

**Figure 1 jcm-11-04065-f001:**
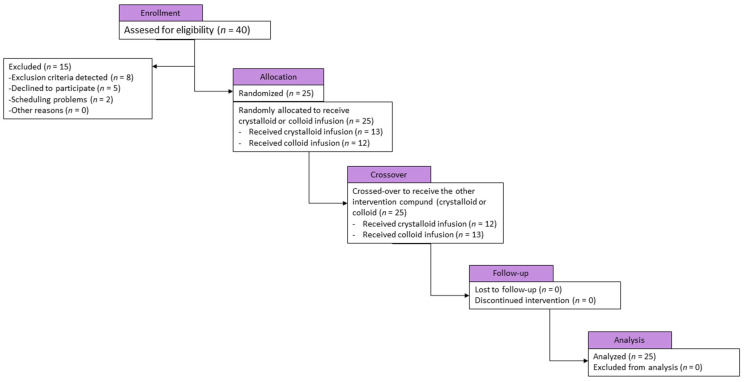
Study flow chart.

**Figure 2 jcm-11-04065-f002:**
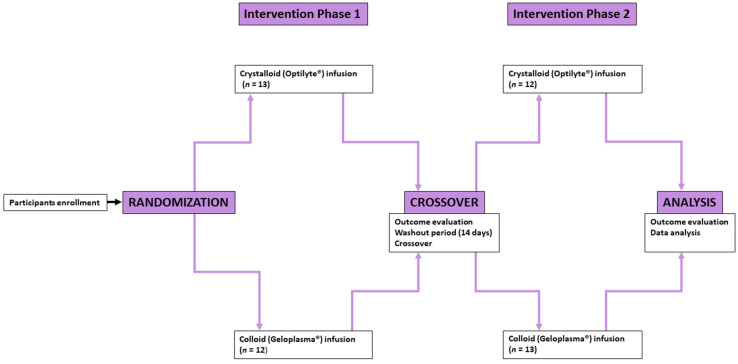
Study protocol.

**Table 1 jcm-11-04065-t001:** Correlations between EXTEM and INTEM clotting times and standard laboratory tests ^1^.

Blood Sample (*n* = 25)	EXTEM Clotting Time		
	PT	Fibrinogen	PLT
Post-dilution_Optilyte	0.53 *p* = 0.007	−0.29 *p* = 0.2	0.03 *p* = 0.9
Post-dilution_Geloplasma	0.46 *p* = 0.019	−0.04 *p* = 0.9	−0.07 *p* = 0.7
**Blood Sample (*n* = 25)**	**INTEM Clotting Time**		
	**APTT**	**Fibrinogen**	**PLT**
Post-dilution_Optilyte	0.714 *p* = 0.0001	0.14 *p* = 0.5	0.04 *p* = 0.9
Post-dilution_Geloplasma	0.644 *p* = 0.0005	0.26 *p* = 0.2	0.11 *p* = 0.6

^1^ Values are Spearman’s coefficients of correlation and their *p*-values. Abbreviations: EXTEM, extrinsic pathway ROTEM assay; INTEM, intrinsic pathway ROTEM assay; APTT, activated partial thromboplastin time; Fibrinogen, fibrinogen concentration; PLT, platelet count; PT, prothrombin time, post-dilution_Optilyte, correlation for coagulation test values post-crystalloid infusion; post-dilution_Geloplasma, correlation for coagulation test values post-colloid infusion.

**Table 2 jcm-11-04065-t002:** Correlations between EXTEM and INTEM alpha angles and standard laboratory tests ^1^.

Blood Sample (*n* = 25)	EXTEM Alpha Angle	
	Fibrinogen	PLT
Post-dilution_Optilyte	0.771 *p* < 0.0001	0.27 *p* = 0.2
Post-dilution_Geloplasma	0.642 *p* = 0.0005	0.31 *p* = 0.1
**Blood Sample (*n* = 25)**	**INTEM Alpha Angle**	
	**Fibrinogen**	**PLT**
Post-dilution_Optilyte	0.495 *p* = 0.012	0.42 *p* = 0.035
Post-dilution_Geloplasma	0.489 *p* = 0.013	0.52 *p* = 0.008

^1^ Values are Spearman’s coefficients of correlation and their *p*-values. Abbreviations: EXTEM, extrinsic pathway ROTEM assay; INTEM, intrinsic pathway ROTEM assay; Fibrinogen, fibrinogen concentration; PLT, platelet count; post-dilution_Optilyte, correlation for coagulation test values post-crystalloid infusion; post-dilution_Geloplasma, correlation for coagulation test values post-colloid infusion.

**Table 3 jcm-11-04065-t003:** Correlations between EXTEM and INTEM clot formation times and standard laboratory tests ^1^.

Blood Sample (*n* = 25)	EXTEM Clot Formation Time	
	Fibrinogen	PLT
Post-dilution_Optilyte	−0.74 *p* < 0.0001	−0.33 *p* = 0.1
Post-dilution_Geloplasma	−0.62 *p* = 0.0009	−0.34 *p* = 0.1
**Blood Sample (*n* = 25)**	**INTEM Clot Formation Time**	
	**Fibrinogen**	**PLT**
Post-dilution_Optilyte	−0.53 *p* = 0.007	−0.46 *p* = 0.02
Post-dilution_Geloplasma	−0.47 *p* = 0.017	−0.57 *p* = 0.003

^1^ Values are Spearman’s coefficients of correlation and their *p*-values. Abbreviations: EXTEM, extrinsic pathway ROTEM assay; INTEM, intrinsic pathway ROTEM assay; Fibrinogen, fibrinogen concentration; PLT, platelet count; post-dilution_Optilyte, correlation for coagulation test values post-crystalloid infusion; post-dilution_Geloplasma, correlation for coagulation test values post-colloid infusion.

**Table 4 jcm-11-04065-t004:** Correlations between EXTEM and INTEM 10th minute clot amplitudes and standard laboratory tests ^1^.

Blood Sample (*n* = 25)	EXTEM Amplitude 10 min		
	Fibrinogen	PLT	D-Dimers
Post-dilution_Optilyte	0.73 *p* < 0.0001	0.34 *p* = 0.1	0.38 *p* = 0.1
Post-dilution_Geloplasma	0.63 *p* = 0.0007	0.33 *p* = 0.1	0.05 *p* = 0.8
**Blood Sample (*n* = 25)**	**INTEM Amplitude 10 min**		
	**Fibrinogen**	**PLT**	**D-Dimers**
Post-dilution_Optilyte	0.63 *p* = 0.0008	0.39 *p* = 0.05	0.36 *p* = 0.1
Post-dilution_Geloplasma	0.59 *p* = 0.002	0.40 *p* = 0.047	0.02 *p* = 0.9

^1^ Values are Spearman’s coefficients of correlation and their *p*-values. Abbreviations: EXTEM, extrinsic pathway ROTEM assay; INTEM, intrinsic pathway ROTEM assay; D-dimer concentration; Fibrinogen, fibrinogen concentration; PLT, platelet count; post-dilution_Optilyte, correlation for coagulation test values post-crystalloid infusion; post-dilution_Geloplasma, correlation for coagulation test values post-colloid infusion.

**Table 5 jcm-11-04065-t005:** Correlations between the EXTEM and INTEM maximum clot firmness and standard laboratory tests ^1^.

Blood Sample (*n* = 25)	EXTEM Maximum Clot Firmness		
	Fibrinogen	PLT	D-Dimers
Post-dilution_Optilyte	0.692 *p* = 0.0001	0.47 *p* = 0.017	0.32 *p* = 0.1
Post-dilution_Geloplasma	0.628 *p* = 0.0008	0.24 *p* = 0.2	0.14 *p* = 0.5
**Blood Sample (*n* = 25)**	**INTEM Maximum Clot Firmness**		
	**Fibrinogen**	**PLT**	**D-Dimers**
Post-dilution_Optilyte	0.67 *p* = 0.0002	0.50 *p* = 0.011	0.41 *p* = 0.039
Post-dilution_Geloplasma	0.599 *p* = 0.002	0.28 *p* = 0.2	0.22 *p* = 0.3

^1^ Values are Spearman’s coefficients of correlation and their *p*-values. Abbreviations: EXTEM, extrinsic pathway ROTEM assay; INTEM, intrinsic pathway ROTEM assay; D-dimer concentration; Fibrinogen, fibrinogen concentration; PLT, platelet count; post-dilution_Optilyte, correlation for coagulation test values post-crystalloid infusion; post-dilution_Geloplasma, correlation for coagulation test values post-colloid infusion.

**Table 6 jcm-11-04065-t006:** Correlations between FIBTEM parameters and standard laboratory tests ^1^.

Blood Sample (*n* = 25)	FIBTEM Maximum Clot Firmness	
	Fibrinogen	D-Dimers
Post-dilution_Optilyte	0.86 *p* < 0.0001	0.47 *p* = 0.02
Post-dilution_Geloplasma	0.87 *p* < 0.0001	0.08 *p* = 0.7
**Blood Sample (*n* = 25)**	**FIBTEM Clotting Time**	
	**Fibrinogen**	**D-Dimers**
Post-dilution_Optilyte	−0.51 *p* = 0.009	−0.26 *p* = 0.2
Post-dilution_Geloplasma	−0.41 *p* = 0.04	0.06 *p* = 0.8

^1^ Values are Spearman’s coefficients of correlation and their *p*-values. Abbreviations: FIBTEM, fibrinogen ROTEM assay; D-dimer concentration; Fibrinogen, fibrinogen concentration; post-dilution_Optilyte, correlation for coagulation test values post-crystalloid infusion; post-dilution_Geloplasma, correlation for coagulation test values post-colloid infusion.

## Data Availability

The datasets used and/or analyzed during the current study are available from the corresponding author upon reasonable request.

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
