# Peer review of "Association between Functional Parameters of Coagulation and Conventional Coagulation Tests in the Setting of Fluid Resuscitation with Balanced Crystalloid or Gelatine: A Secondary Analysis of an In Vivo Prospective Randomized Crossover Study"

_jcm, 2022, doi:10.3390/jcm11144065_

Round 1
Reviewer 1 Report
The manuscript by Wiorek et al. reported associations between coagulation parameters measured with conventional coagulation tests and rotational thromboelastometry (ROTEM). This study was a secondary analysis of a recently published prospective study in which subjects received Optilyte and Geloplasma infusion in a randomized, crossover manner over a 14 day time span. The goal of the study was to describe associations between standard and functional parameters of coagulation and fibrinolysis in the setting of acute hemodilution.
The rationale and study design are adequately described in the introduction and methods sections. However, the Results section simply restates the data shown in the Tables without further interpretation or contextualization, making it challenging to decipher/understand the meaning of the data. This is a major limitation of the manuscript that should be improved.
Major Comments:
- The title should reflect that this is a secondary analysis
- Figure 3 is a graphical presentation of data previously reported in the article published in 2022 in J. Pers. Med. This figure should be removed to avoid duplication of data.
- Why are the correlations at the pre-dilution timepoint different for the two groups. Can you better explain what these differences mean? And, what does a lack of correlation in the pre-diluted colloid group imply for using ROTEM vs conventional assays for non-hemodiluted patients?
- Why is the pre-dilution data even reported given that goal was to assess the correlations in hemodiluted patients? If there is some added value in knowing the correlations for the pre-diluted samples, then this should be more clearly stated.
- A more granular discussion of the data with regards to the physiology underlying the coagulation parameters for EXTEM vs INTEM vs FIBTEM would be helpful for contextualizing the findings.
- The conclusion that ROTEM assays “sufficiently” correlate with conventional coagulation test is too subjective. How do you define “sufficiently?” Perhaps there is a more quantitative assessment that can define the overall strength of the correlations.
- Some discussion as to why differences were observed following infusion with crystalloids vs colloids is warranted.
Author Response
Reviewer #1
We would like to kindly thank you for your extensive review, substantive approach, and valuable comments and observations concerning our study. Thank you for your approval of its rationale and design adequacy. Thank you also for your comments and observations regarding the results. We performed corrections to the Results and Tables by removing the duplication of correlation coefficient values and corrected the overall manuscript following your suggestions that we’re hoping would meet your expectations.
- Thank you kindly for your observation. We clarified the title of our manuscript, including the information about it being a secondary analysis.
- Thank you for this observation. We acknowledge this graphical presentation may fall under the category of data duplication; therefore, we removed Figure 3.
- Thank you for this observation. To be honest, it’s difficult to find a direct answer to this interesting question. The issue of connections between conventional coagulation tests and functional coagulation tests in the setting of clinical utilization among various patients has not been exhausted in the available literature and is still open for investigation. The lack of correlations in healthy patients may arise from factors unrelated to hemostasis. Is this an impact of demographics or other subject-related factors? We will definitely put further efforts into our studies continuation in a larger population.
- Thank you for this vital remark concerning the clarity of our study’s purpose. As the pre-dilution data do not provide additional information, may be somehow misleading, and may impede the study interpretation, to avoid unnecessary confusion, we decided to remove the additional pre-dilutional data to make clearer to the reader about the actual objective of the study. The post-dilution values have been adjusted to the degree of hemodilution.
- Thank you for your concern regarding the contextualization of our study findings. To improve this purpose, we added a brief description to the „Methods” section of the manuscript about the basis of the utilized ROTEM assays and their role in the assessment of the specific stage of the physiologic coagulation process.
- To objectify our study conclusion, we followed your suggestion and replaced the subjective wording of „sufficiently correlated” with methodologically proper statistical terminology, as the observed correlations ranged from „moderate” to „strong”.
- Thank you for the suggestion regarding the discussion of the differences following infusion with crystalloids vs colloids. We complied with your comment and added this needed point in our manuscript's „Discussion” section regarding the potentially expected different volume effect of crystalloids and colloids, along with some additional cited sources regarding the issue.

Reviewer 2 Report
Authors performed the randomized study on healthy volunteers, they described the in vivo association between standard and functional parameters of coagulation and fibrinolysis in the setting of acute hemodilution induced by an infusion of balanced crystalloid or synthetic gelatine solutions. They hypothesized that independently of the infused fluid type and degree of induced haemodilution, the results of ROTEM are well correlated with the classic coagulation studies.
This manuscript is potentially interesting, but there are many unclear points and authors explanations are unsatisfactory.
1) The objective is not clear.
2) The explanations of methods are not satisfactory.
3) The conclusion is also not clear.
4) The explanations for EXTEM and INTEM are required. What is FIBTEM?
5) There are many tables. These may be able to be delated or combined.
6) All tables need further explanations in legends.
7) Standard laboratory tests are required.
Author Response
Reviewer #2
We would like to kindly thank you for reviewing our manuscript. Thank you for your comments and observations. We are glad you’ve found interest in the conducted research. We followed through with your suggestions. Please find below our responses to your comments and suggestions and a summary of implemented revisions and corrections.
- To clarify the objective of our study, we added a further explanation in the „Introduction” section of our manuscript, where we present our rationale for investigation. We aimed to describe the in vivo association between standard and functional parameters of coagulation and fibrinolysis, that is, to evaluate the statistical correlation between these parameters taking into account the degree of dilution associated with the fluid infusion, in the setting of acute hemodilution induced by an infusion of balanced crystalloid or synthetic gelatine solutions.
- Thank you for this remark. As this study is a secondary analysis, the detailed description of methods has been previously described and published. We referenced the full-text version of our recently published work with full specification and methodology, which we couldn’t replicate in this manuscript due to the risk of data duplication and auto-plagiarism, which was a critical comment of Reviewer #1. The full methodology description is also included in the ClinicalTrials.gov database, as referenced in the manuscript.
- We complied with your comment regarding the clarity of the study conclusion. We applied corrections to the wording of the conclusion, that will hopefully improve its reception.
- Thank you kindly for your inquiry. We followed through with this suggestion and referenced further source material to refer the insightful and astute reader to in order to provide any further explanations required.
- Thank you for this observation. Of course, we clarified this issue and modified the tables as suggested. We aimed to make our tables clearer to the reader by removing the accessory information concerning the pre-dilutional data that did not contribute particularly to our study's actual objective.
- As you suggested, we included further explanations in the tables’ legends regarding the terminology contained in them.
- We acknowledge your suggestion. As we’ve previously mentioned, we needed to avoid the potential of data duplication and the risk of auto-plagiarism, as we clarified in the study description that this is a secondary analysis. Therefore we could not include the entirety of this data and values due to the risk of duplication of the previously published results and the overall risk of exceeding the maximum allowed volume of the manuscript. However, to make it clearer to the reader, we included further reference to the more detailed results in our previously published manuscript, including the values of standard laboratory tests.

Round 2
Reviewer 1 Report
The authors have sufficiently addressed my comments/critiques
Reviewer 2 Report
Authors sufficiently responded to the comments. I have no further comment.